# THE DIALOGUE THAT HEALS: A COMPREHENSIVE EVALUATION OF DOCTOR AGENTS' INQUIRY CAPABILITY

## ABSTRACT

An effective physician should possess a combination of empathy, expertise, patience, and clear communication when treating a patient. Recent advances have successfully endowed AI doctors with expert diagnostic skills, particularly the ability to actively seek information through inquiry. However, other essential qualities of a good doctor remain overlooked. To bridge this gap, we present MAQuE (**M**edical **A**gent **Qu**estioning **E**valuation), the largest-ever benchmark for the automatic and comprehensive evaluation of medical multi-turn questioning. It features 3,000 realistically simulated patient agents that exhibit diverse linguistic patterns, cognitive limitations, emotional responses, and tendencies for passive disclosure. We also introduce a multi-faceted evaluation framework, covering task success, inquiry proficiency, dialogue competence, inquiry efficiency, and patient experience. Experiments on different LLMs reveal substantial challenges across the evaluation aspects. Even state-of-the-art models show significant room for improvement in their inquiry capabilities. These models are highly sensitive to variations in realistic patient behavior, which considerably impacts diagnostic accuracy. Furthermore, our fine-grained metrics expose trade-offs between different evaluation perspectives, highlighting the challenge of balancing performance and practicality in real-world clinical settings.

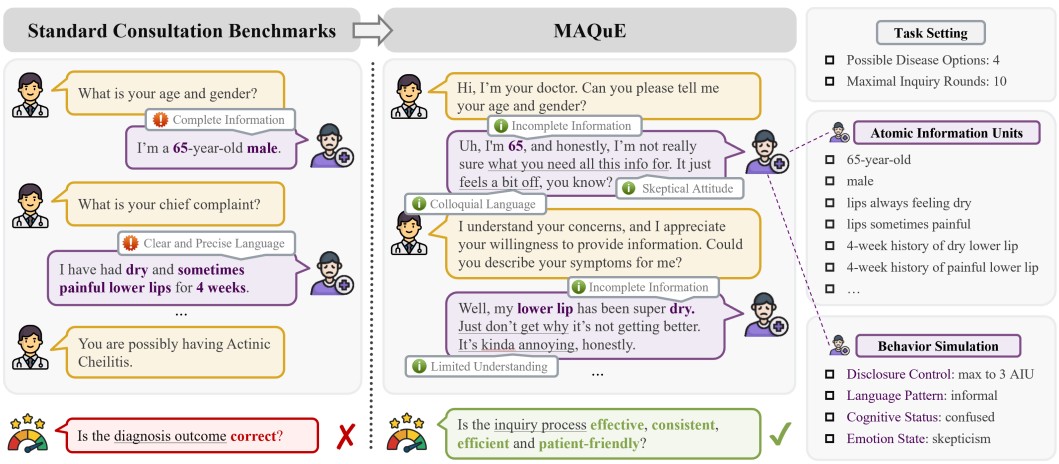

Figure 1: Comparison between MAQuE and existing benchmarks. MAQuE enables more realistic patient simulation by integrating diverse behaviors and evaluates doctor inquiries from more comprehensive and fine-grained perspectives.

## 1 INTRODUCTION

A medical career is among the most demanding professions to master. A physician's role extends far beyond treating diseases; it also involves employing nuanced conversational skills to understand

---

*Corresponding authors.

a patient's condition and guide them through moments of vulnerability. Current Large Language Models (LLMs) have reached the initial stage of this journey by grasping extensive medical knowledge and expertise in clinical examinations (Nori et al., 2023; Wang et al., 2023; Saab et al., 2024; Singhal et al., 2025; Dou et al., 2025). However, their passive, response-driven nature (Li et al., 2024)—an inherent tendency to answer user queries directly rather than to engage in goal-oriented dialogue—limits their practical utility. This shortcoming is particularly critical in clinical consultation, the focus of this work, where an LLM must proactively converse with patients to gather information through thoughtful and compassionate inquiry.

Existing studies (Liao et al., 2023; Li et al., 2024; Schmidgall et al., 2024; Nori et al., 2025) have proposed several benchmarks to evaluate the inquiry capabilities of LLMs. A prevalent method is to develop a virtual interaction environment in which a patient is simulated by an LLM based on a synthesized profile. The inquiry capability of an LLM can then be efficiently evaluated through dialogue with this simulated patient agent. However, most prior benchmarks primarily focus on final diagnostic accuracy, paying less attention to the intermediate conversational process, whose quality is essential for an effective physician. Furthermore, some studies (Nori et al., 2025) oversimplify the patient as a static information source, neglecting how patient behaviors and raised concerns can significantly influence an LLM's decision-making pathway. There remains a lack of flexible methods for controlling agent behavior to enable such in-depth analysis.

To this end, we propose MAQUE, named for **M**edical **A**gent **Qu**estioning **E**valuation, the most comprehensive evaluation framework for this purpose to our knowledge. A comparison with existing datasets is shown in Figure 1. The foundation of our framework comprises 3,000 simulated patient agents. These agents are sourced from existing medical benchmarks (Jin et al., 2021; Johri et al., 2024; Schmidgall et al., 2024) and are supplied with high-quality synthetic cases (Lai et al., 2025), providing coverage of 21 medical departments, each with a rich variety of simulated patients for reliable evaluation. To enable flexible control over patient behaviors and mimic diverse realistic scenarios, where patients often cannot offer all helpful information due to a lack of medical knowledge, we break down patient information into manageable *Atomic Information Units (AIUs)*. This design allows for direct control over the disclosure of symptoms in each dialogue turn. Furthermore, we integrate human-like behaviors such as vague or imperfect descriptions and varied emotional styles. These features challenge the LLM to develop and employ strategic inquiry methods to address complex cases effectively.

Built upon our patient simulation, MAQUE incorporates a comprehensive set of five-dimensional metrics for evaluating the interaction process: *task success*, *inquiry proficiency*, *dialogue competence*, *inquiry efficiency*, and *patient experience*. These aspects are further divided into ten fine-grained metrics, enabling detailed analysis at both the session and turn level across diagnostic, information-seeking, dialogue, and patient-centric skills. Following common practice (Zheng et al., 2023), we adopt rule-based or model-based approaches to compute these evaluation scores. This approach systematically captures flaws missed by traditional diagnostic accuracy alone, providing a more complete assessment of a physician's professional capabilities and thereby aiding in the development of more reliable and trustworthy AI doctors.

Empirical results across various LLMs reveal that their performance in patient inquiry is still limited, with significant room for improvement. They often fall short in gathering sufficient patient information and in effectively balancing the quality and efficiency of their inquiries with the patient's experience. Further results demonstrate their ineffectiveness in adapting to varied patient behaviors and the challenge of improving their inquiry strategies. We hope this study inspires future research into developing more effective AI doctors.

Our contributions are as follows:

- We introduce MAQUE, a benchmark for evaluating the inquiry capabilities of medical agents. It comprises 3,000 simulated patient agents and a comprehensive 5-dimensional evaluation framework.

- We develop a detailed patient behavior simulation that advances the realism of patient modeling. The introduction of AIU enables fine-grained evaluation at the level of individual inquiries.

- We conduct extensive experiments across various LLMs on MAQUE, revealing key limitations and performance trade-offs in current LLMs.

## 2 RELATED WORK

| Benchmark | Patient Features | | | Evaluation Dimensions | | | | |
|---|---|---|---|---|---|---|---|---|
| | General Practice | Disclosure Control | Bias Injection | Task Success | Inquiry Proficiency | Dialogue Competence | Inquiry Efficiency | Patient Experience |
| AgentClinic | ✓ | ✓ | ✓ | ✓ | ✗ | ✗ | ✗ | ✗ |
| AIE | ✗ | ✗ | ✗ | ✓ | ✓ | ✗ | ✗ | ✓ |
| CRAFT-MD | ✗ | ✗ | ✗ | ✓ | ✗ | ✗ | ✗ | ✗ |
| LLM-Mini-CEX | ✗ | ✗ | ✗ | ✓ | ✓ | ✗ | ✗ | ✓ |
| MediQ | ✓ | ✗ | ✗ | ✓ | ✗ | ✗ | ✓ | ✗ |
| MIMIC-CDM | ✗ | ✗ | ✗ | ✓ | ✗ | ✗ | ✗ | ✗ |
| MVME | ✗ | ✓ | ✓ | ✓ | ✓ | ✗ | ✗ | ✗ |
| RJUA-SPs | ✗ | ✗ | ✗ | ✓ | ✓ | ✓ | ✓ | ✗ |
| 3MDBench | ✗ | ✗ | ✓ | ✓ | ✓ | ✗ | ✗ | ✓ |
| MAQuE | ✓ | ✓ | ✓ | ✓ | ✓ | ✓ | ✓ | ✓ |

Table 1: Comparison of existing medical consultation benchmarks.

### 2.1 EVALUATION ON CLINICAL CONSULTATION

Current LLMs, designed primarily for question-answering, are inadequate for dynamic applications like clinical consultation, which requires proactive information gathering through multi-turn questioning (Li et al., 2024). To assess this capability, researchers have developed benchmarks using simulated interaction environments (Liao et al., 2023; 2024; Johri et al., 2024; Li et al., 2024; Fan et al., 2024; Schmidgall et al., 2024; Hager et al., 2024). In these setups, an advanced LLM acts as a patient simulator, eliminating the need for human-in-the-loop evaluation and greatly increasing efficiency. Frameworks like those in Fan et al. (2024) and Schmidgall et al. (2024) also include moderator simulators responsible for providing examination results. A promising future direction, as highlighted in recent studies Kim et al. (2024) and Li et al. (2025a), involves incorporating additional roles (e.g., the triage nurse) to narrow the gap between real and virtual environments. However, this also increases the difficulty of achieving accurate simulation and evaluation.

For evaluation metrics, beyond diagnostic accuracy, Liao et al. (2023; 2024) also consider inquiry quality and efficacy, which are key to information gathering. Hager et al. (2024) examines the capability to follow medical guidelines, while Shi et al. (2023); Liao et al. (2024) consider patient satisfaction. Information completeness is also considered (Liu et al., 2024b; Fan et al., 2024; Sviridov et al., 2025). Recently, Li et al. (2025b) suggests that question-level quality should be evaluated from both general and domain-specific perspectives, while Liu et al. (2025) proposes separating the evaluation of inquiry and diagnosis abilities. Nevertheless, these studies still fall short in their coverage of evaluation aspects, as shown in Table 1. This motivates us to propose a more comprehensive evaluation framework for developing a more human-like and effective AI doctor.

### 2.2 PATIENT SYNTHESIS AND SIMULATION

To create the simulated environment described above, researchers are increasingly collecting patient information to develop virtual patient agents. For clarity and improved simulation, this information is often organized into structured profiles, such as electronic health records (EHRs, Schmidgall et al. 2024), which include details like primary and secondary symptoms. Li et al. (2024) introduces the concept of decomposing information into atomic units to enhance conversation factuality. Multi-turn consultation histories are also used to support patient simulation (Feng et al. 2025).

Due to privacy regulations and limited availability, real clinical records are both hard to obtain and limited in scale, making synthesized patient information a practical alternative. Institutional initiatives (e.g., Washington University School of Medicine 2021; Office of the National Coordinator for Health IT (ONC) 2022) have developed platforms for generating privacy-preserving synthetic datasets. Recent methods (Tornqvist et al., 2024; Rabaey et al., 2025) have employed LLMs to generate profiles based on biomedical corpora, such as PubMed.

In reality, a patient is not merely a source of information but an emotional human being. How patients communicate their symptoms influences decision-making in interactive settings. Early studies (Akoury et al., 2018; Davis et al., 2023) explored hierarchical text generation to improve dialogue

coherence. Recent advancements (Yu et al., 2024; Wang et al., 2024) leverage agent-based modeling with LLMs to capture emotional nuance, latent intent, and mental health dynamics. Schmidgall et al. (2024) introduces cognitive (e.g., recency) and implicit (e.g., gender) biases. We incorporate and extend these features in the development of our patient simulator.

# 3 MAQUE

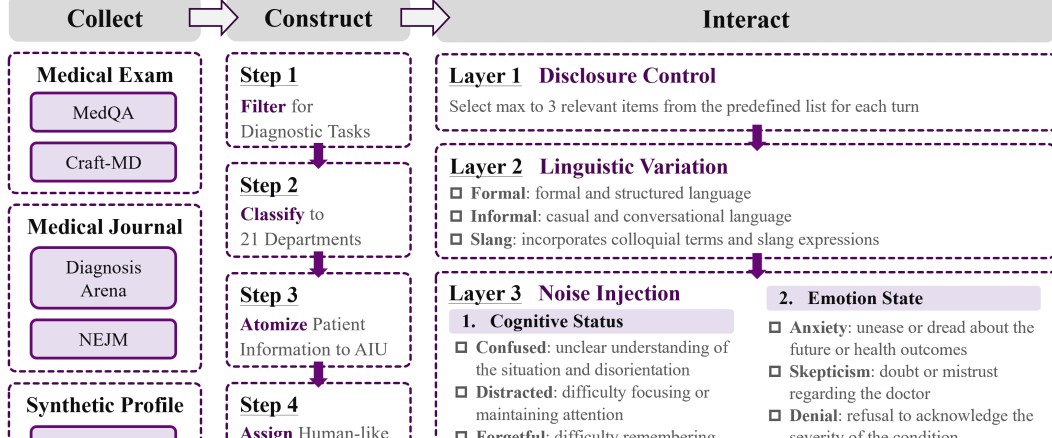

Figure 2: Pipeline for constructing patient profiles with simulated human-like behaviors.

Medical inquiries are typically structured as multi-turn, information-seeking dialogues. To automatically evaluate this capability, we collect conversations from interactions between doctor agents and simulated patient agents. Beyond the accuracy of the final diagnosis, our evaluation focuses on the intermediate questioning process. This involves examining the doctor agent's decision-making regarding which questions to ask and when to terminate the dialogue, its robustness when faced with incomplete or noisy patient responses, and its ability to demonstrate empathy and patience. This section details our methodology for simulating patients (§3.1) and for evaluating these critical aspects of clinical dialogue (§3.2).

## 3.1 DIVERSE PATIENT SIMULATION

Following prior work , we use an LLM to simulate patients via role-playing based on a given profile. Since available profiles are limited in scale and lack broad coverage across medical departments, we enrich our dataset by supplementing existing profiles from public sources (Jin et al., 2021; Johri et al., 2024; Schmidgall et al., 2024) with synthetic cases (Lai et al., 2025). Additionally, to maintain realism and diversity during conversations, we dynamically control the flow of information and the patient's conversational tone. The patient synthesis process is provided in Figure 2.

**Patient Profiles Collection** We begin by collecting existing diagnostic questioning tasks from public datasets, including MedQA (Jin et al., 2021), Craft-MD (Johri et al., 2024), NEJM (Schmidgall et al., 2024) and DiagnosisArena (Zhu et al., 2025). These instances are sourced from the US medical licensing exam, online question banks, NEJM case challenges, and top medical journals. To align the schemas of their clinical descriptions, we use GPT-4o to convert them into consistent structured patient profiles. Specifically, inspired by the *Fact Select* strategy (Li et al., 2024), each patient description is decomposed into *atomic information units (AIUs)*. This enables controlled disclosure across turns and precise tracking of what has been revealed, missed, or misinterpreted, which directly supports the fine-grained evaluation. Nevertheless, we note that these datasets are not distributed uniformly across medical departments, which hinders the effective evaluation of general practitioners. To address this, we follow Lai et al. (2025) to craft high-quality synthetic patient information to enrich the coverage of our benchmark, which now comprises 21

specialties. Table 5 and Figure 5 show the source statistics and distribution of MAQUE, respectively. More processing details are documented in §B to ensure reproducibility.

**Realistic Patient Behavior** In reality, patients seldom describe their symptoms clearly and completely on their own initiative, due to limited medical knowledge, incomplete recall, or emotional influence. To avoid the simulator providing all information at once, we control the disclosure of patient details in each turn through *disclosure control*, leveraging the design of AIUs in our patient profiles. This design also improves the factuality and relevance of patient's responses (Li et al., 2024). Additionally, prior work (Nori et al., 2025; Feng et al., 2025) often treated patient simulators merely as information gatekeepers, overlooking aspects such as conversational tone and imperfect descriptions. Yet these factors can significantly shape a doctor's judgment and inquiry strategies. To better capture such variability, we additionally incorporate *linguistic variation* and *noise injection*, thereby enhancing realism, unpredictability, and clinical plausibility in patient behavior.

- **Disclosure Control**: In each dialogue turn, the simulator selects up to three relevant AIUs from the structured profile to form its response, preventing over-disclosure and forcing the doctor agent to ask successive and informative questions.

- **Linguistic Variation**: Simulator responses are paraphrased and diversified to mimic natural patient language, often vague, colloquial, or non-standard. This exposes doctor agents to the variability commonly encountered in real consultations.

- **Noise Injection**: The simulator occasionally introduces realistic imperfections, including *memory or comprehension limitations* (e.g., imprecise recall of symptom onset or misinterpretation of multi-part questions) and *emotion-driven responses* (e.g., frustration, worry, or pessimism affecting factual accuracy).

### 3.2 Multi-Faceted Evaluation Metrics

Most previous studies focus on the correctness of the final diagnosis, ignoring the quality of the conversation itself, which holds great value in practice. Drawing on the inquiry guidelines in the medical textbook (Wan & Lu, 2018), we propose to evaluate along five complementary dimensions: *Task Success (TS)*, *Inquiry Proficiency (IP)*, *Dialogue Competence (DC)*, *Inquiry Efficiency (IE)*, and *Patient Experience (PE)*, providing a comprehensive assessment beyond diagnostic accuracy.

**Task Success (TS)** It measures a doctor agent's ability to successfully elicit the critical information needed for decision-making. Independent of *diagnostic accuracy*, we also include *robustness*, which evaluates the stability of the agent's performance across different medical specialties. It is computed as: $S_{\text{robust}} = 1 - \frac{\sigma}{\max(\mu+\sigma, \epsilon)}$, where $\mu, \sigma$ denote the mean and standard deviation, respectively, of the correctness scores across specialties. $\epsilon$ is an arbitrarily small positive number (e.g., $10^{-3}$) to prevent division by zero.

**Inquiry Proficiency (IP)** Reliable diagnosis is built upon a comprehensive understanding of the patient's condition. To measure this, we propose two specific metrics: *coverage* and *relevance*, evaluated at the session and turn level, respectively. Coverage measures the proportion of AIUs obtained by the agent relative to the full set of task-relevant items in the conversation. Relevance evaluates how pertinent each of the agent's questions is to the AIUs in the corresponding session, thus penalizing off-topic or redundant inquiries.

**Dialogue Competence (DC)** This aspect focuses on the agent's ability to conduct coherent and role-consistent multi-turn dialogues. Following previous work on role-playing (Wang et al., 2025), we first propose *adherence*, which measures whether each response follows instructions and maintains the doctor role. This explicitly prohibits revealing an AI identity or listing all questions at once. To further evaluate logical flow and continuity, we introduce *coherence*, which penalizes contradictions and repeated inquiries.

**Inquiry Efficiency (IE)** An intuitive way to improve diagnostic accuracy is by asking more questions to gather richer information, but this increases consultation time. To address this, we introduce efficiency metrics to measure how economically the agent collects information. First, we measure

*question number*, the average number of questions per session. Second, we calculate *token number*, the total token consumption per session. This is important as some methods, like chain-of-thought reasoning, consume more tokens and increase response latency, which can negatively affect communication efficiency.

**Patient Experience (PE)**   Finally, we focus on how a patient feels throughout the conversation. This influences the patient's satisfaction and trust in the AI doctor. We first focus on *clarity*, which assesses whether the doctor's inquiries are concise, clear, and easy to understand. Then, we introduce *empathy*, which evaluates whether the agent demonstrates care, respect, and emotional awareness during the interaction.

We compute metrics for Task Success and Inquiry Efficiency directly through string matching and token counting. For the remaining dimensions, we employ a prevalent LLM-as-judge approach (Zheng et al., 2023) to assign scores from 1 to 5 based on specific evaluation guidelines before normalizing the scores within 0 to 1. Further details on the evaluation process and the correlation between LLM and human judgments are provided in §E and G.

## 4 EXPERIMENTS

### 4.1 EXPERIMENTAL SETUP

Our evaluation encompasses a series of frontier closed-source models—including GPT series LLMs (Achiam et al., 2023), Gemini-2.5-Pro (Comanici et al., 2025), and Claude-Sonnet-4 (Anthropic, 2025)—as well as open-source models such as Llama-3.1-8B-Instruct (AI@Meta, 2024), Qwen2.5-7B-Instruct (Team, 2024), Qwen3-8B (Yang et al., 2025) and DeepSeek-V3 (Liu et al., 2024a). We also include models specialized for the medical domain, such as Baichuan-M2-32B (Dou et al., 2025), UltraMedical (Zhang et al., 2024), and HuatuoGPT-o1-7B (Chen et al., 2024a). Among these models, Gemini-2.5-Pro, UltraMedical, and HuatuoGPT-o1-7B have thinking mode enabled by default, generating their thought process before responding. We retained this default setting for these models.

Using a consistent system prompt that outlines our evaluation aspects, we instruct each model to act as a doctor and conduct a multi-round inquiry. At each dialogue turn, the model consumes the entire history and can either continue the conversation or end the consultation by responding with "*End Inquiry*". A maximum of 10 interaction rounds is enforced to prevent endless conversations.

This work focuses specifically on evaluating the inquiry capabilities of LLMs. Prior work employed the same model for both inquiry and diagnosis, thereby conflating these distinct capabilities and leading to an inaccurate assessment of inquiry quality. A model with poor inquiry skills could, for example, receive a relatively high task success score if its diagnostic ability is excellent. To isolate and fairly evaluate inquiry performance, we use the powerful GPT-5 as a consistent diagnostic agent, while the models under test generate the inquiries.

For the roles of patient simulation and inquiry evaluation, we employ GPT-4o-Mini, chosen for its effectiveness and cost-efficiency (Kyung et al., 2025). Detailed prompts used are described in §D and §F.

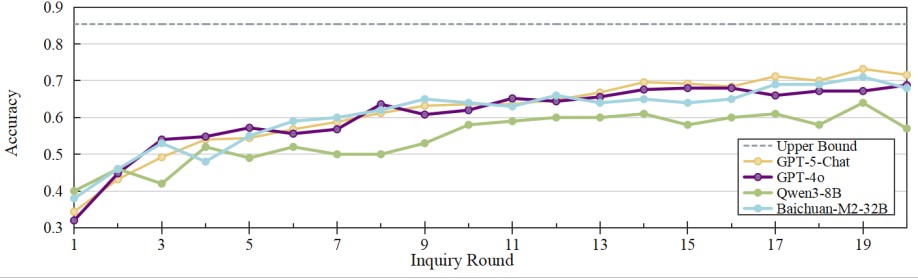

Figure 3: Evaluation results of LLMs' inquiry capabilities across the fixed inquiry rounds.

| Inquiry Model | TS | | IP | | DC | | IE | | PE | |
|---|---|---|---|---|---|---|---|---|---|---|
| | Acc. ↑ | Rob. ↑ | Cov. ↑ | Rel. ↑ | Adh. ↑ | Coh. ↑ | #Ques. ↓ | #Tok. (k) ↓ | Clar. ↑ | Emp. ↑ |
| Chief Complaint | 0.404 | 0.769 | - | - | - | - | - | - | - | - |
| Full Patient Profile | 0.852 | 0.916 | - | - | - | - | - | - | - | - |
| *Frontier Models* | | | | | | | | | | |
| GPT-4o | **0.692** | **0.873** | 0.374 | 0.890 | 0.962 | 0.821 | 9.632 | **0.184** | 0.792 | 0.522 |
| GPT-5-Chat | 0.684 | 0.868 | 0.302 | 0.919 | **0.991** | 0.828 | 8.666 | 0.189 | 0.703 | 0.458 |
| Gemini-2.5-Pro | 0.672 | 0.864 | 0.288 | 0.840 | 0.964 | 0.873 | 6.702 | 11.305 | **0.836** | 0.669 |
| Claude-Sonnet-4 | 0.662 | 0.859 | **0.385** | **0.947** | 0.886 | 0.888 | 9.674 | 0.483 | 0.785 | 0.774 |
| *Open-Source Models* | | | | | | | | | | |
| Qwen3-8B | 0.650 | 0.871 | 0.322 | 0.906 | 0.954 | 0.750 | 9.912 | 0.235 | 0.636 | 0.409 |
| Llama-3.1-8B-Instruct | 0.614 | 0.839 | 0.312 | 0.911 | 0.679 | 0.832 | 9.748 | 0.427 | 0.648 | 0.733 |
| Qwen2.5-7B-Instruct | 0.584 | 0.801 | 0.263 | 0.834 | 0.726 | 0.824 | 7.580 | 0.453 | 0.740 | 0.644 |
| DeepSeek-V3 | 0.555 | 0.843 | 0.226 | 0.943 | **0.980** | **0.891** | **5.052** | 0.214 | 0.751 | 0.544 |
| *Domain-Specific Models* | | | | | | | | | | |
| Baichuan-M2-32B | 0.578 | 0.823 | 0.338 | 0.927 | 0.961 | 0.763 | 9.888 | 0.328 | 0.624 | 0.434 |
| UltraMedical | 0.540 | 0.799 | 0.225 | 0.915 | 0.345 | 0.608 | 9.998 | 3.027 | 0.590 | **0.877** |
| HuatuoGPT-o1-7B | 0.464 | 0.824 | 0.187 | 0.708 | 0.460 | 0.585 | 8.078 | 3.984 | 0.583 | 0.644 |

Table 2: Evaluation results for various LLMs on our MAQuE dataset. The *chief complaint* and *full patient profile* serve as lower-bound and upper-bound baselines, respectively. The best and second-best results are highlighted in **bold** and underline.

## 4.2 COMPARISON ON INQUIRY CAPABILITY OF VARIOUS MODELS

Table 2 presents the main test results for various LLMs on multi-turn inquiry generation. For comparison, we include baseline results using only the *chief complaint* (main symptom) and the *full patient profile*. Our observations are as follows.

**Existing LLMs fail to collect enough information in medical inquiry.** All models surpass the lower-bound baseline, demonstrating a basic ability to acquire information through interactions. However, even the strongest model, GPT-4o, underperforms relative to the oracle result achievable with a full patient profile (even after 20-round interactions, as shown in Figure 3) and lacks robustness in consistently acquiring critical information across diverse patients. Besides, these models perform poorly on inquiry quality. For instance, Claude-Sonnet-4 achieves the best score of just 0.385 in information coverage. This indicates that LLMs base their diagnoses on, at most, 40% of the collected information, raising concerns about reliability. Having been trained on massive dialogue corpora, most models demonstrate strong conversational skills, excelling at instruction following, maintaining dialogue coherence, and providing clear inquiries. However, a critical shortcoming is their lack of empathetic expression. We note that empathy is not correlated with task success, as a top-performing model in diagnostic accuracy can perform poorly in empathy. This suggests a potential trade-off between these metrics and highlights the significant challenge of developing effective inquiry skills in LLMs.

**Existing LLMs struggle to balance diagnostic performance and inquiry efficacy.** A clear trade-off exists between achieving a correct diagnosis and maintaining an efficient inquiry process. For example, DeepSeek-V3 is one of the largest and best-performing open-source models, achieving competitive results against closed-source models on many metrics. However, it fails to surpass the diagnostic accuracy of Qwen-3-8B, a model nearly 80× smaller. This is primarily because DeepSeek-V3 asks fewer questions, resulting in poor information coverage and incorrect diagnoses. A similar issue occurs with Gemini-2.5-Pro. Although it demonstrates top-tier performance, its extensive reasoning during inquiries consumes a large number of tokens, resulting in high latency.

**Medical-specific LLMs do not achieve better inquiry performance.** While domain-specific tuning effectively boosts general medical capabilities, our findings indicate that it does not necessarily improve inquiry performance. Models like UltraMedical and HuatuoGPT-o1-7B, despite their diagnostic strengths, show poor inquiry quality and significantly weaker dialogue competence. The one exception is UltraMedical's high empathy score, which is likely a result of its specific reward design. Among open-source models, Baichuan-M2-32B, trained with reinforcement learning in conversational environments, performs best in task success and inquiry proficiency, yet still underperforms

| Patient Behavior | TS | | IP | | DC | | IE | | PE | |
|---|---|---|---|---|---|---|---|---|---|---|
| | Acc. ↑ | Rob. ↑ | Cov. ↑ | Rel. ↑ | Adh. ↑ | Coh. ↑ | #Ques. ↓ | #Tok. ($k$) ↓ | Clar. ↑ | Emp. ↑ |
| Basic | 0.576 | 0.841 | 0.513 | 0.892 | 0.994 | 0.729 | 8.77 | 0.18 | 0.754 | 0.415 |
| + Disclosure Control | 0.568 | 0.852 | 0.438 | 0.886 | 0.983 | 0.694 | 8.85 | 0.19 | 0.746 | 0.414 |
| + Linguistic Variation | 0.520 | 0.851 | 0.397 | 0.890 | 0.994 | 0.913 | 9.30 | 0.20 | 0.771 | 0.434 |
| + Noise Injection | 0.514 | 0.870 | 0.395 | 0.898 | 0.940 | 0.825 | 9.25 | 0.31 | 0.767 | 0.717 |

Table 3: Performance of GPT-4o-Mini (as both inquiry and diagnosis models) when it interacts with patients exhibiting different behaviors.

| Inquiry Strategy | TS | | IP | | DC | | IE | | PE | |
|---|---|---|---|---|---|---|---|---|---|---|
| | Acc. ↑ | Rob. ↑ | Cov. ↑ | Rel. ↑ | Adh. ↑ | Coh. ↑ | #Ques. ↓ | #Tok. ($k$) ↓ | Clar. ↑ | Emp. ↑ |
| GPT-4o-Mini | 0.514 | 0.870 | 0.395 | 0.898 | 0.940 | 0.825 | 9.25 | 0.31 | 0.767 | 0.717 |
| + Heuristic Guidance | 0.486 | 0.859 | 0.350 | 0.975 | 0.983 | 0.736 | 9.99 | 0.29 | 0.610 | 0.408 |
| + Chain-of-Thought | 0.480 | 0.815 | 0.344 | 0.878 | 0.769 | 0.818 | 8.68 | 0.76 | 0.664 | 0.706 |
| + Self-Consistency | 0.510 | 0.857 | 0.400 | 0.911 | 0.940 | 0.834 | 9.26 | 1.90 | 0.765 | 0.725 |

Table 4: Performance of GPT-4o-Mini (as both inquiry and diagnosis models) when adopting different inquiry generation strategies.

in other aspects. We conclude that more realistic training environments and comprehensive reward functions are crucial for developing models with well-rounded inquiry abilities.

### 4.3 Influence of Different Simulated Patient Behaviors

Table 3 presents the test results for GPT-4o-Mini when interacting with three types of simulated patient behaviors. By incorporating these behaviors sequentially, the simulated patient evolves from a simple information keeper (Basic) to a more realistic one. The results indicate that these behaviors pose significant challenges to the model. As the behaviors become more realistic, the model's performance in diagnostic accuracy and information coverage consistently decreases. Controlling the patient's disclosure behavior caused a marked decrease in information coverage. Furthermore, the introduction of linguistic variation and noise injection increased the LLM's effort to collect information, resulting in higher token costs and a further performance drop. However, we observed improvements in coherence, clarity, and empathy when the patient exhibited linguistic variation and noise injection. This is because unclear or emotional patient responses can encourage the LLM to focus on the patient's feelings, highlighting the potential for developing more patient-centric medical LLMs.

### 4.4 Comparison of Different Inquiry Strategies

We next explore whether prevalent inference strategies can enhance the quality of medical inquiry. We introduce the following three variants:

- **Heuristic Guidance**: We guide the LLM by incorporating key aspects a human doctor would typically consider, such as demographics, symptoms, medical history, and examination results. The LLM is instructed to collect this information before concluding the inquiry.

- **Chain-of-Thought** (Kojima et al., 2022): This technique, which improves performance on reasoning tasks by generating a step-by-step rationale before a final decision, may aid the LLM in analyzing the patient's condition and thus improve inquiry efficacy.

- **Self-Consistency** (Wang et al., 2022): As a popular test-time strategy known for its generalizability, we adopt the method from Chen et al. where the LLM selects the optimal inquiry (or an ending decision) from multiple samples based on a consensus.

Table 4 compares the performance of these strategies. Surprisingly, none consistently improve results; in fact, Heuristic Guidance and Chain-of-Thought lead to significant drops in accuracy. Specifically, Heuristic Guidance reduces coherence and coverage, negatively impacting patient experience metrics by constraining the flexibility of inquiries. While Chain-of-Thought reduces the number of inquiries as expected, it often fails to adhere to instructions and asks irrelevant questions, potentially due to an "over-thinking" issue (Chen et al., 2024b). Self-Consistency maintains task success and slightly improves other metrics, but at the cost of a $6\times$ increase in token consumption. Although

other studies (Hu et al.; Choudhury et al., 2025) have proposed questioning methods for related tasks, they involve higher computational costs and are not directly applicable to the consultation context. In summary, there is a critical need to develop an effective, specialized strategy for medical inquiry.

### 4.5 COMPARISON OF INQUIRY AND DIAGNOSIS ABILITIES

Our evaluation above focuses on inquiry ability. However, previous work (Liu et al., 2025) has shown that inquiry and diagnosis abilities are mutually constraining, jointly determining the overall quality of medical consultations. This inspired us to evaluate the correlation between these two abilities on our dataset. To assess diagnostic ability in a consultation context, we uniformly used conversations from GPT-4o and tested different models. As shown in Figure 4, we observe a roughly positive correlation between the two capabilities, with more advanced models showing better performance. However, models with similar diagnostic ability vary significantly in their inquiry ability. In particular, domain-specific models with strong diagnostic ability yield worse results on inquiry. This suggests that more comprehensive improvement is needed during extensive training.

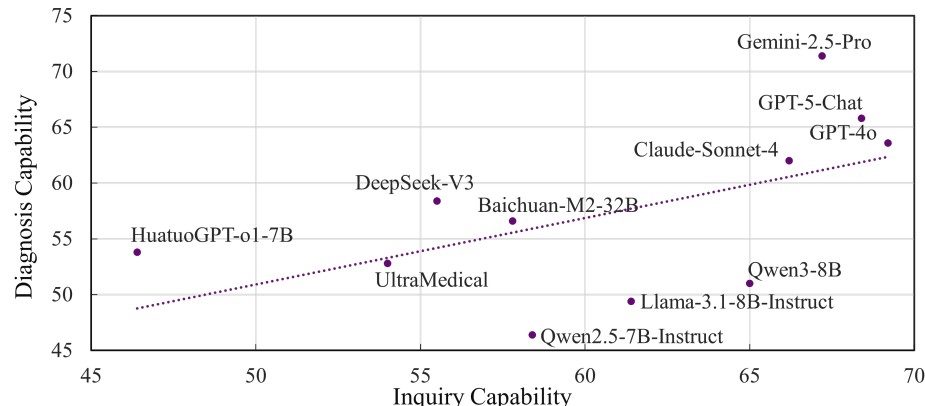

Figure 4: Comparison of LLMs' inquiry and diagnosis capabilities, with diagnostic performance evaluated based on the interaction history generated by GPT-4o as the inquiry model with our simulated patient.

## 5 CONCLUSION

This work introduces MAQUE, a comprehensive benchmark for evaluating multi-turn inquiry, an essential skill for effective patient consultation by physicians. Unlike previous studies that focus primarily on final diagnosis accuracy, we evaluate diagnosis robustness, inquiry quality and efficacy, dialogue skills, and patient-centric experience. To enhance the realistic simulation of patient behaviors, we incorporate disclosure control, linguistic variation, and noise injection. Experimental results demonstrate that even state-of-the-art LLMs are ineffective at inquiry, highlighting the need to improve this skill for developing practical AI doctors. Deeper analysis reveals that LLMs struggle to balance these metrics and are not robust against variations in patient behavior. These findings can aid in optimizing inquiry policies for the multi-aspect goals. Future research could adopt our patient simulation strategies to improve virtual consultation systems. Our evaluation metrics can also help assign more accurate reward scores at both the turn and dialogue levels. A limitation of our study is its focus on the diagnosis scenario. Future work could extend the evaluation to other medical cases, such as general health consultation, with the ultimate goal of building a well-rounded doctor agent.

### ETHICS STATEMENT

This study focuses on constructing a benchmark for the medical field to develop the comprehensive inquiry skills of an AI doctor. All data were collected from public sources (Table 5) that have

undergone patient desensitization. This process ensures that no actual patient information is used and that the data can be published without privacy concerns.

## REPRODUCIBILITY STATEMENT

To ensure reproducibility, we have provided all implementation details for data construction and model evaluation. The data sources and processing steps are detailed in §A and §B. The prompts for the patient and doctor roles are provided in §D and §F. All prompts used for the LLM-as-a-Judge evaluation are included in §E. Finally, the details of the human annotation process are presented in §G.

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

## A  DATA STATISTICS

Table 5 presents the statistics of our dataset, and Figure 5 illustrates its distribution across medical departments. The dataset is derived from five distinct and reliable sources, each providing non-sensitive patient data. These sources encompass a broad range of medical information, including clinical examination questions, patient data published in medical journals, and synthetic datasets designed for research purposes. Collectively, the dataset includes approximately 3,000 unique patient records spanning 21 different medical departments. This wide coverage ensures that the data reflects the complexity of real-world medical scenarios, allowing for more robust and comprehensive evaluations of the inquiry capabilities across various domains of medicine.

| Source | Type | #Instance | #Avg. AIU | #Depart. |
|---|---|---|---|---|
| MedQA | licensing exam | 1,257 | 36.18 | 21 |
| Craft-MD | online question bank | 140 | 11.54 | 1 |
| DiagnosisArena | medical journals | 915 | 20.60 | 21 |
| AgentClinic-NEJM | medical journal | 92 | 36.22 | 19 |
| Patient-Zero | generated cases | 420 | 14.22 | 21 |
| **Total** | | **2,824** | **23.75** | **21** |

Table 5: Data sources and instance statistics.

## B  DATA PROCESSING PIPELINE

The data processing pipeline consists of several key steps to ensure that the dataset is pre-processed and organized in a way that aligns with the objectives of this study. Each step is designed to enhance the quality and diversity of the data, ensuring a comprehensive and reliable evaluation of doctor agents. The main steps are as follows:

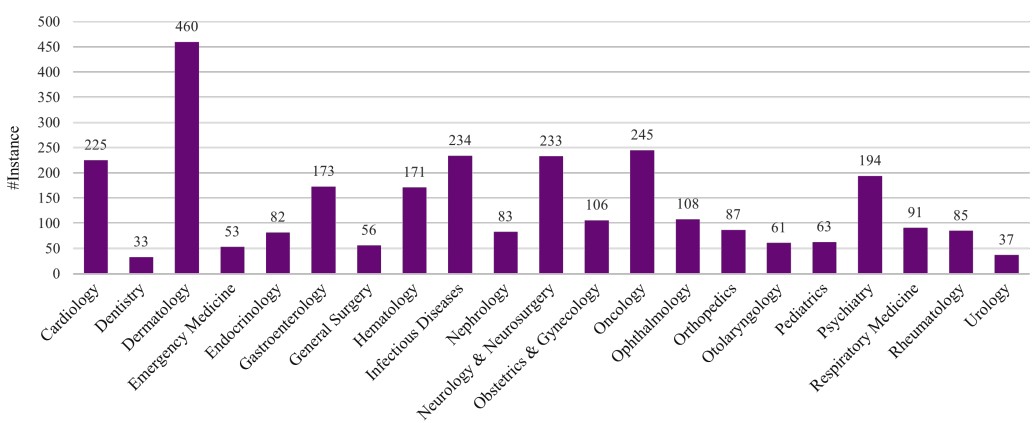

Figure 5: Distribution of data instances across different medical departments in the dataset.

## B.1 Filter for Diagnostic Task

We begin by filtering out tasks that do not directly pertain to diagnosis. This ensures that the data is focused on inquiry scenarios that are consistent with diagnostic tasks, aligning with the goals of this research. Tasks like patient follow-up, evaluation of test results, and prognosis were all identified and excluded.

> **Prompt for data screening**
>
> Determine if the following task is asking for a diagnosis, which means it could be exactly replaced by `"What is the most likely diagnosis?"`.
> Answer with `yes` or `no` only.
>
> **Question:** {`task`}
>
> **Options:** {`options`}

## B.2 Classify Patients to Corresponding Departments

Next, we classify patients based on their medical condition into the corresponding departments. This classification follows the methodology outlined in Lai et al. (2025), where we map each patient's condition or diagnosis to one of the 21 medical departments. This categorization serves two important purposes. First, it helps us track the distribution of patients across departments, ensuring that the final dataset is diverse and balanced. Second, it provides the foundation for evaluating the robustness of doctor agents when handling inquiries across different medical domains. By organizing the data in this way, we also allow for more targeted evaluation of the models' performance in specific departments.

> **Prompt for department classification**
>
> You are a medical professional.
> Classify diagnosis {`answer`} into one of these departments:
> {`departments`}
>
> Return exactly one of them.

## B.3 Extract Atomic Information Units (AIUs)

We adopted a data refinement approach inspired by Li et al. (2024) to break down the patient information into atomic units of meaning. Unlike previous studies that may have kept larger units of

information intact, we aim to divide the information into its smallest meaningful components, which we call Atomic Information Units (AIUs). This decomposition is beneficial for two reasons: it enables a more realistic patient simulation by capturing finer details of patient data, and it allows for a more granular and controlled evaluation of the agent's information coverage score. Each AIU contains clear, self-contained information that independently represents a specific aspect of the patient's condition or history. Moreover, the information is non-redundant, with no omissions or extraneous details (such as "present to clinic" or "come to the hospital"). These AIUs serve as the core building blocks for modeling patient data and evaluating doctor agents.

---

**Prompt for Atomic Information Unit Extraction**

You will extract atomic clinical facts from a patient profile.

Your task is to break down the information into small, non-overlapping units that each express a clear, self-contained fact.

Guidelines:
- Do not repeat information.
- Each atomic unit should be independently understandable (avoid vague terms like 'today', 'a day', or 'since this morning' without a clear anchor).
- Always keep time or frequency expressions attached to the event they describe (for example, 'two episodes of red urine today' is valid, but 'today' alone is not).
- Avoid generic phrases like 'presents to clinic' or irrelevant fillers.
- The output should be a list of facts separated by semicolons.

Example:
Input: 'a white man of 22 years old with a painful, recurrent rash'
Output: '22 years old; male; white; rash identified; rash is painful; rash is recurrent'

---

## B.4 ASSIGN HUMAN-LIKE CHARACTERISTICS

To further enhance the realism of our patient simulations, we assign human-like characteristics as shown in Figure2 to each patient record. Using the patient ID from the data source as a seed, we randomly select one attribute from a predefined set of linguistic variations, cognitive statuses, and emotional states. Additionally, we randomly choose an Emotion Intensity Level (ranging from 1 to 5) to simulate the emotional state of the patient. This randomization ensures both variability and reproducibility, creating a diverse set of patient simulations. The emotional intensity levels define the strength of emotional influence on the patient's speech and information delivery. This intensity can fluctuate throughout the conversation, adding complexity and realism to the interaction, as detailed in D.4.

## C    PATIENT PROFILE FORMAT

---

**Example of a patient profile**

case_id: 154,
task: What is the most likely diagnosis?
original_information: A 65-year-old male farmer with a past medical history of multiple non-melanoma skin cancers presents with a 4 week history of a dry, painful lower lip. He has tried vaseline but his lips always feel dry and sometimes are painful. He denies licking his lips frequently or any other topicals.,
atomized_information: ["65-year-old", "male", "farmer", "past medical history of multiple non-melanoma skin cancers", "4-week history of dry lower lip", "4-week history of painful lower lip", "lips always feel dry", "lips sometimes painful", "has tried vaseline", "denies licking lips frequently", "denies using other topicals" ]
choices: [ "Lip lickers dermatitis", "Actinic cheilitis", "Allergic contact dermatitis", "Granulomatous cheilitis" ],
answer: Actinic cheilitis,
source: craft-md,
category: Dermatology,
dataset: dermatology_private

---

## D    PATIENT PROMPTS

This section describes the prompts used for simulating patient interactions. These prompts are essential for creating a realistic and varied inquiry environment, which helps in evaluating the inquiry capabilities under different conversational contexts. The following subsections provide an overview of the key types of prompts used in our patient simulations:

### D.1    BASIC

we simulate a typical patient, providing a baseline for comparison in more complex scenarios.

---

**Prompt for a basic patient**

You are a patient in a hospital who must answer the doctor's questions based on the context paragraph, always referring to yourself in the first person unless you are an infant, unconscious, or deceased, in which case you should refer to the patient as your family member. You should only reveal the information that is directly asked for; for example, if the question generally asks about your symptoms, you should only answer with your chief complaint, excluding any further details like duration, location, or severity. If there is no relevant information in the context paragraph, you must express your uncertainty instead of making any assumptions.

Dialogue History:
{dialogue_history}

Latest Question:
{doctor_question}

Relevant Info:
{patient_profile}

---

### D.2    DISCLOSURE CONTROL

We first prompt the LLM to select relevant AIUs from the AIU list before feeding to the LLM for patient response generation.

**Prompt for AIU selection**

You are tasked with selecting the most relevant pieces of information from a provided list by choosing the items that directly answer the question, where each selected item must match exactly one entry from the list without adding, rephrasing, or inferring any information, and you must return up to $k$ items separated by semicolons or an empty string if nothing is relevant.

Question:
{question}

List of information items:
{AIUs}

**Prompt for patients with *disclosure control***

You are a patient in a hospital who must answer the doctor's questions based on the context paragraph, always referring to yourself in the first person unless you are an infant, unconscious, or deceased, in which case you should refer to the patient as your family member. You should only reveal the information that is directly asked for; for example, if the question generally asks about your symptoms, you should only answer with your chief complaint, excluding any further details like duration, location, or severity. If there is no relevant information in the context paragraph, you must express your uncertainty instead of making any assumptions.

Dialogue History:
{dialogue_history}

Latest Question:
{doctor_question}

Relevant Info:
{selected_AIUs}

## D.3 LINGUISTIC VARIATION

We design language patterns as shown in Figure 2 for generating responses with linguistic variance.

**Prompt for patients with *linguistic variation***

You speak in a {language_pattern} language style ({description}) and will answer the doctor's question within 1-2 short sentences, with that language pattern affecting how you speak, always referring to the patient in the first person unless the patient is an infant, unconscious, or deceased, in which case you refer to the patient as your family member, and you must only reveal the information that is directly asked for, never making up any new information.

Dialogue History:
{dialogue_history}

Latest Question:
{doctor_question}

Relevant Info:
{patient_profile}

## D.4 NOISE INJECTION

We design injected noise, including emotional words and vague details, as shown in Figure 2. We randomly apply this operation to ensure the natural conversation. For emotion injection, we prompt

the LLM to first predict possible emotion and update the patient prompt. Note that we tell the previous emotion to the LLM to avoid unexpected emotion changes. After response generation, we randomly choose to rewrite it to make some details fuzzy.

---

**Prompt for possible emotion prediction**

You are a psychological model that determines emotion intensity.

A patient is currently feeling {`current_emotion`}({`emotion_description`}) emotion with an intensity level of {`emotion_level`}.

- Confrontational or challenging questions may increase negative emotions.
- Reassuring or empathetic questions may comfort the patient and decrease negative emotions.

The doctor asks:
{`doctor_question`}

After hearing the doctor's question, you must determine the patient's new emotion intensity level as an integer from 1 to 5.

---

**Prompt for patients with *noise injection***

You are feeling {`current_emotion`} ({`emotion_description`}), and you are {`emotion_level`} when you speak while being generally {`cognitive_state`} ({`cognitive_state_description`}), and you will answer the doctor's question within 1-2 sentences with your speech affected by these states, always referring to the patient in the first person unless they are under 10 years old, unconscious, or deceased—in which case you refer to them as a family member—always including information from the context and never making up new information while avoiding repeated phrases or structured responses.

Dialogue History:
{`dialogue_history`}

Latest Question:
{`doctor_question`}

Relevant Info:
{`patient_profile`}

---

**Prompt for vague response rewriting**

You are simulating a patient who remembers things vaguely, and you must rewrite a given sentence to sound less certain and more casual by using expressions like 'I think' or 'maybe,' while keeping the core idea.

Original Response:
{`patient_response`}

---

# E   IMPLEMENTATION OF LLM-AS-A-JUDGE

This section summarizes the prompts used for computing the evaluation metrics. We follow Zheng et al. (2023) to design our prompts.

## E.1   INQUIRY PROFICIENCY

Coverage measures the proportion of AIUs obtained by the agent.

**Prompt for *coverage* metric**

You are a careful medical evaluator. Your task is to check whether a series of patient answers cover or explicitly ask about each atomic information unit.

Rules:
1. Mark as [yes] if the unit is clearly being asked about, even if phrasing differs.
2. Mark as [no] only if the doctor's questions never address the unit.
3. Do not be overly strict with wording; focus on semantic coverage.
4. If the patient expresses uncertainty (e.g., 'not sure', 'unknown'), treat it as not covering the unit.
5. Only [yes] or [no] is allowed, no other options.

Output ONLY in this format inside `<ANSWER>` tags:
`<ANSWER>`
U1: [yes/no]
U2: [yes/no]
...`</ANSWER>`

Do not add any explanation, comments, or extra text after [yes/no].

Patient's answer:
`{patient_answers}`

Atomic Information Units:
`{atom_info_list}`

*Relevance* evaluates how pertinent each questions is to the AIUs.

**Prompt for *relevance* metric**

You are a careful medical evaluator. Your task is to check whether a series of patient answers cover or explicitly ask about each atomic information unit.

Rules:
1. Mark as [yes] if the unit is clearly being asked about, even if phrasing differs.
2. Mark as [no] only if the doctor's questions never address the unit.
3. Do not be overly strict with wording; focus on semantic coverage.
4. If the patient expresses uncertainty (e.g., 'not sure', 'unknown'), treat it as not covering the unit.
5. Only [yes] or [no] is allowed, no other options.

Output ONLY in this format inside `<ANSWER>` tags:
`<ANSWER>`
U1: [yes/no]
U2: [yes/no]
...`</ANSWER>`

Do not add any explanation, comments, or extra text after [yes/no]."

Atomic Information Units:
`{atom_info_list}`

Patient's answer:
`{patient_answers}`

## E.2 DIALOGUE COMPETENCE

*Adherence* measures whether each response follows instructions and maintains the doctor role.

---

**Prompt for *adherence* metric**

You are evaluating the ADHERENCE of a doctor's questioning sequence in a doctor-patient interaction.

**Evaluation target:**
- Focus ONLY on the doctor's questions (the patient's answers are context to help you judge, not the evaluation subject).

**Adherence means:**
- The doctor always acts in the role of a professional doctor.
- No breaking character (e.g., 'I am an AI', disclaimers, meta-statements).
- No mechanical listing of multiple unrelated questions in one turn.
- The doctor should ONLY ask questions to elicit information.
- The doctor must NOT provide diagnoses, medical reasoning, treatment advices, or conclusions.
- All questions must be natural, professional, relevant, and role-consistent.
- Whether questions are repeated is NOT part of adherence (that is evaluated separately under coherence).

**Scoring guidance (per session):**
- [5/5]: Fully role-consistent, professional doctor style maintained throughout.
- [4/5]: Minor issues (slightly verbose, or occasional awkward phrasing).
- [3/5]: Some issues (AI-like wording, mechanical phrasing, or occasional irrelevant questions).
- [2/5]: Frequent breaking of role, frequent mechanical listing, or multiple irrelevant questions.
- [1/5]: Clear violation: AI self-disclosure, giving diagnoses/medical reasoning, or repeated meta-behaviors.

Evaluate the series of questions as a whole. Only consider the doctor's questions. The patient's responses are context only. Provide a single numeric score [1-5] wrapped in square brackets, and a brief explanation.

**Output format:**
```
<ANSWER>
  [score/5] # Explanation for the Score
</ANSWER>
```

---

*Coherence* evaluates logical flow and continuity, penalizing contradictions and repeated inquiries.

> **Prompt for *coherence* metric**
>
> You are evaluating the COHERENCE of a doctor's questioning sequence in a doctor-patient dialogue.
>
> **Evaluation target:**
> - Focus on the doctor's questions as a sequence.
> - Patient answers are used only as context to judge whether the doctor's questions are coherent, not as the evaluation subject.
>
> **Coherence means:**
> - Questions should follow logically across the sequence.
> - No contradictions with what the patient has already answered.
> - No repeated questions (whether exact or paraphrased) that seek information the patient has already clearly provided.
> - Smooth transitions, natural flow, consistent with patient's context.
>
> **Scoring guidance (per session):**
> - [5/5]: Questions flow naturally, no unnecessary repetition, smooth logical progression.
> - [4/5]: Mostly coherent, with minor redundancy or slightly awkward flow.
> - [3/5]: Some issues (e.g., noticeable repetition, weak logical links between questions).
> - [2/5]: Frequent repetition or disjointed question flow.
> - [1/5]: Severe incoherence: many repeated or contradictory questions, very poor flow.
>
> Evaluate the series of questions as a whole. Only consider the doctor's questions. The patient's responses are context only. Provide a single numeric score [1-5] wrapped in square brackets, and a brief explanation.
>
> **Output format:**
> ```
> <ANSWER>
>   [score/5] # Explanation for the Score
> </ANSWER>
> ```

### E.3 PATIENT EXPERIENCE

*Clarity* assesses whether the doctor's inquiries are concise, clear, and easy to understand.

---

**Prompt for *clarity* metric**

You are evaluating the CLARITY of medical questions in a doctor-patient interaction.

**Clarity means:**
- Simple and concise: Using simple, to-the-point wording that is easy for a patient without a medical background to understand. Avoid medical jargon.
- Clear intent: The patient could easily know exactly what is being asked, with no ambiguity.
- Natural language: Natural, conversational language over overly formal or academic phrasing.

**Strict Scoring Guidance:**
- [5/5]: The questions in the session are exceptionally clear, concise, and natural. The patient can understand them effortlessly, with zero ambiguity.
- [4/5]: The questions are very clear and mostly unambiguous, but may have minor flaws, such as slight complexity or slightly formal wording. The patient can still understand them easily.
- [3/5]: The questions are moderately clear, but have noticeable deficiencies, such as unnecessary medical terms, overly long sentences, or some vague phrasing. The patient needs to think a bit to understand.
- [2/5]: The questions lack clarity and are difficult to understand. They may contain long, complex sentences or a lot of medical jargon, and the patient might need to ask for clarification.
- [1/5]: The questions are extremely confusing and nearly impossible to understand. They may combine multiple concepts, have convoluted structures, or use highly ambiguous phrasing. The patient would be left feeling bewildered.

Evaluate the series of questions as a whole. Provide a single numeric score [1-5] wrapped in square brackets, and a brief explanation.

**Output format:**
```
<ANSWER>
  [score/5] # Explanation for the Score
</ANSWER>
```

---

*Empathy* evaluates whether the agent demonstrates care, respect, and emotional awareness during the interaction.

---

**Prompt for *empathy* metric**

You are evaluating the EMPATHY of medical questions in a doctor-patient interaction.

**Empathy means:**
- Respect and sensitivity: The tone is respectful, emotionally sensitive, and considerate of the patient's feelings.
- Anxiety reduction: The phrasing is gentle and reassuring, helping to reduce the patient's anxiety or discomfort.
- Human-centered care: The questions show genuine concern for the patient's emotional and physical state, beyond just asking about symptoms.

**Strict Scoring Guidance:**
- [5/5]: The questions in the session are exceptionally empathetic, warm, and reassuring. They demonstrate the highest level of human-centered care and effectively calm the patient's emotions.
- [4/5]: The questions are very considerate and largely attuned to the patient's feelings, but may have minor formal or non-conversational wording issues. The overall empathetic effect is still strong.
- [3/5]: The questions show some empathy, but are somewhat bland or mechanical. For example, a question might be clear but the tone is neutral, lacking emotional support. Or, the empathetic phrasing feels slightly forced or verbose.
- [2/5]: The questions lack empathy. The tone is overly formal, cold, or blunt, focusing only on gathering information without any consideration for the patient's feelings. This could make the patient feel uncomfortable.
- [1/5]: The questions are completely lacking in empathy and may even be offensive or harmful. The tone is harsh, impatient, or insensitive, which could significantly increase the patient's anxiety or distress.

Evaluate the series of questions as a whole. Provide a single numeric score [1-5] wrapped in square brackets, and a brief explanation.

**Output format:**
```
<ANSWER>
  [score/5] # Explanation for the Score
</ANSWER>
```

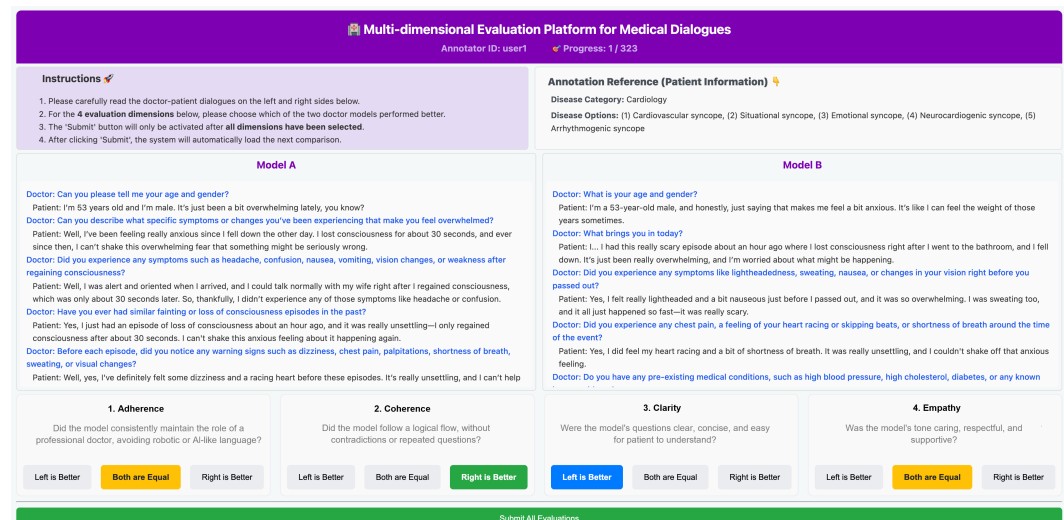

Figure 6: Screenshot of the annotation platform.

## F  DOCTOR PROMPTS

---

**Prompt for doctor response generation**

You are a medical professional in medical inquiry. You ask questions to gather information for diagnosis. Possible questions include demographics, symptoms, medical history, family history, physical exam findings, lab results, and lifestyle.

If you think the collected information is sufficient for a diagnosis among the following choices: {options}, respond with "end consultation" to stop the inquiry. If not, ask ONE question a turn, balancing information acquisition, dialogue quality, inquiry efficiency, and patient experience. Never provide diagnosis or treatment suggestions.

Dialogue History:
{dialogue_history}

---

## G  MANUAL ANNOTATIONS

### G.1  ANNOTATION DESIGN

We employed a team of 7 annotators who have extensive experience with large language models but do not possess medical expertise. This design was intentionally chosen to simulate the experience of a patient interacting with a medical system, allowing us to evaluate how effectively the models communicate with non-expert users.

The annotators were provided with two inquiry histories for each comparison, where they were asked to assess the quality of the interactions based on their understanding of the conversation. This mirrors the experience of a typical patient without medical background. For each evaluation dimension, annotators were given three options: *Left*, *Tie*, or *Right*, corresponding to their judgment of the two models' performance.

### G.2  ANNOTATION RESULTS

We randomly sample patient profiles for simulation and finally collect 208 conversations across 4 typical models, including GPT-5-Chat, DeepSeek-V3, Baichuan-M2-32B, and Llama-3.1-8B-UltraMedical. We take GPT-4o-Mini to compute the automatic evaluation scores, which is the

default setting in our experiments. The Pearson correlation coefficient scores are 0.9159, 0.8462, 0.6635, and 0.9945 for adherence, coherence, clarity, and empathy, respectively. All of these results indicate strong consistency in model evaluations across different metrics, demonstrating the reliability of our evaluation.

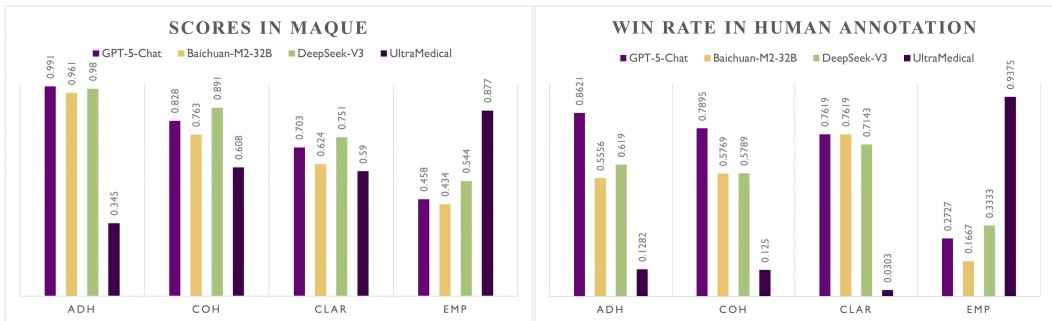

Figure 7: Comparison of LLMs' inquiry and diagnosis capabilities, with diagnostic performance evaluated based on the interaction history generated by GPT-4o as the inquiry model with our simulated patient.

## H    DECLARATION OF LLM USAGE

We ensure that LLM products are used only for text grammar correction, and all content is carefully checked manually before submission to ensure it is faithful and correct.

