# OpenReview forum: "The Dialogue That Heals: A Comprehensive Evaluation of Doctor Agent's Inquiry Capability"
_ICLR.cc/2026/Conference — ICLR 2026 Conference Withdrawn Submission_

### Official Review · Reviewer_qZfd · 2025-10-26

**Soundness:** 3
**Presentation:** 3
**Contribution:** 3
**Rating:** 6
**Confidence:** 4

**Summary:**

This work proposes an evaluation framework for the medical inquiry capabilities of LLMs. Conceptually, it is different from many existing works in terms of what information is included and what is evaluated. Empirically, it shows that medical inquiry is still a challenging problem.

**Strengths:**

+ medical inquiry is an important research question
+ the resource, when released, will be helpful

**Weaknesses:**

- It's nice to try to disentangle the inquiry and diagnosis capabilities in medical applications. In line 307, the authors use "the powerful GPT-5" as the diagnosis part and test models on the inquiry part. GPT-5 certainly isn't perfect: I wonder for the models that are more distilled from GPT-5-generated texts (which we don't know), would their outputs/inquiries be more compatible with GPT-5 as a diagnosis agent? Essentially, this could be a confounder. I wonder if any chance the diagnosis part could be replaced with an oracle diagnosis, limited in sample size and with human diagnosis perhaps, to remove the confounder and see if the patterns hold.

- Figure 3 seems to be important info about the multi-turn skills: it would be nice to have open models in the mix.

- I like that on line 374 the authors show that existing medical LMs are not doing so well on the proposed evaluation framework, thanks to its more "real-world"ness. Medical NLP is really an interesting domain: every paper always highlights how previous papers are very "un-real-world" and how we became more "real-world", while we never seem to be "real-world" enough. For the authors, if there's something about this work that's not "real-world" enough, what would that be? Will we ever get a data/evaluation that most people could agree is "real-world" enough?

- Figure 4 got me thinking: can we somehow have different models responsible for different steps? For example, one model is good at inqury do inquiry, one model is good at diagnosis do diagnosis. Or if the authors believe there should still be one model doing the best on everything.

**Questions:**

please see above

---

> ### Author Response · Authors · 2025-11-24
>
> Thank you for recognizing our work! Now, we will address each of your comments and concerns one by one.
>
> ---
>
> **W1: Diagnostic Agent**
>
> Thank you for your insightful suggestion. In response, we have added a set of experiments where we cross-combine different inquiry and diagnosis models to observe variations in the results:
>
> | Diagnosis model \ Inquiry model | GPT-5 | GPT-5-Chat | GPT-4o | Qwen3-8B | Baichuan-M2-32B |
> | ------------------------------- | ----- | ---------- | ------ | -------- | --------------- |
> | GPT-5                           | 0.54  | 0.67       | 0.66   | 0.62     | 0.59            |
> | GPT-5-Chat                      | 0.55  | 0.62       | 0.61   | 0.56     | 0.57            |
> | GPT-4o                          | 0.43  | 0.46       | 0.36   | 0.32     | 0.38            |
> | Qwen3-8B                        | 0.39  | 0.46       | 0.38   | 0.42     | 0.4             |
> | Baichuan-M2-32B                 | 0.48  | 0.54       | 0.46   | 0.47     | 0.44            |
>
> - The ranking of inquiry performance is consistent with the trend observed in Table 2.
> - GPT-5 consistently demonstrates the best diagnostic performance, supporting its suitability as a stable diagnostic agent, while its inquiry performance is not the highest, indicating that its reasoning module is not well-suited for the inquiry task.
>
> ---
> **W2: Open Models for Figure 3**
>
> Thank you for your constructive suggestion! We have tested Qwen3-8B and Baichuan-M2-32B in the fixed-round setting and will update Figure 3. Both models converge near 20 rounds, and there is still a significant gap from the upper bound(0.854).
> | Inquiry Round      | 2     | 4     | 6     | 8     | 10    | 12    | 14    | 16    | 18    | 20    |
> |-----------------|-------|-------|-------|-------|-------|-------|-------|-------|-------|-------|
> | Qwen-3-8B       | 0.425 | 0.485 | 0.52  | 0.535 | 0.56  | 0.605 | 0.605 | 0.645 | 0.645 | 0.635 |
> | Baichuan-M2-32B | 0.5   | 0.5   | 0.54  | 0.52  | 0.54  | 0.64  | 0.7   | 0.64  | 0.68  | 0.6   |
>
>
> ---
> **W3: Real-World Limitation**
>
> Thank you for your insightful question. Our work enhances realism through a three-layer patient simulation (Fig. 2) and a five-aspect evaluation framework (Sec. 3.2). These components are designed to align with the real clinical inquiry process (Wan & Lu, 2018, Diagnosis, 9th ed., pp. 77-78). However, some bias persists due to the inherent complexity of patient-doctor interactions. As noted in line 485, we believe that beyond diagnosis, an AI doctor should also be tested on general health consultation. Multi-modal interaction is another key consideration, where a doctor can observe a symptom directly rather than relying only on the patient's description. In summary, our study is a step toward a more realistic setup, but we acknowledge there is still a long way to go.
>
> ---
> **W4: Model Cooperation**
>
> That is an interesting idea! We agree that combining different models for inquiry and diagnosis has the potential to build stronger agent systems. This is also related to the concept of MoE or model routing([1][2]), where specialized models collaborate and complement one another across different stages of a process. We will provide further discussion on this in the revision.
> > [1] Wensheng Gan et al. Mixture of Experts (MoE): A Big Data Perspective. arXiv preprint arXiv:2501.16352, 2025.
> > [2] Wittawat Jitkrittum et al. Universal Model Routing for Efficient LLM Inference. arXiv preprint arXiv:2502.08773, 2025.

---

### Official Review · Reviewer_uTeh · 2025-10-31

**Soundness:** 3
**Presentation:** 3
**Contribution:** 2
**Rating:** 4
**Confidence:** 4

**Summary:**

This paper presents an evaluation framework for medical agents, tackling different aspects of a conversation. A 3,000-sample dataset, MAQUE, is created for this purpose, incorporating diverse linguistic patterns, cognitive limitations, emotional responses, and tendencies for passive disclosure to simulate real-world environments. The framework includes five evaluation dimensions to assess agents' capabilities comprehensively. The authors conduct extensive experiments across a wide range of LLMs, and the results reveal that these models still fall short of the requirements for actual clinical settings.

**Strengths:**

1. A dataset containing 3,000 samples is constructed for medical case evaluation, which is sufficient for comprehensive assessment.

2. An evaluation framework is also proposed, covering a wide range of dimensions. Instead of simply focusing on lexical overlap, this paper emphasizes the semantic aspects of medical conversations.

3. A wide range of LLMs are evaluated, and several interesting findings are presented for reference:

- LLMs cannot compete with physicians currently.

- LLMs require long context to achieve good performance.

- Medical LLMs do not yield better performance than general LLMs.

**Weaknesses:**

1. It seems this paper only involves simulated patients and doctors. Despite including non-expert human evaluation, it is difficult to assess the real-world performance of these LLMs.

2. LLMs can be easily affected by variations in context. However, no case studies are conducted to illustrate how they are affected. It would be beneficial to see more detailed analysis on this aspect. Given the strong text understanding capabilities of LLMs, I would expect these variations to stem from challenging cases.

3. Line 357: "This indicates that LLMs base their diagnoses on, at most, 40% of the collected information, raising concerns about reliability." I believe most people, as well as physicians, make judgments based on partial information. Additionally, this should not serve as evidence to support "Existing LLMs are far from being effective physicians," because the LLMs' performance is compared against an upper bound rather than actual physicians.

**Questions:**

1. Since the dataset is simulated, have you conducted correlation analysis between expert evaluations and the proposed metrics? It would be beneficial to include such analysis as a reference for validation.

2. Additionally, how do these LLMs perform with real patients? Do they demonstrate consistent performance across simulated and real-world scenarios?

---

> ### Author Response · Authors · 2025-11-24
>
> Thank you for your thoughtful comments and feedback! We will respond to your comments and questions one by one.
>
> ---
>
> **W1: Limited real-world applicability**
>
> Please note that this research aims to create a simulated environment for evaluation. While real-world performance is the most reliable metric, conducting such an evaluation is often time-consuming, costly, and raises privacy concerns, which can prohibit this line of research. As shown in Table 2, various studies have focused on developing simulated evaluation environments. We build upon this work by considering more comprehensive metrics that reflect real-world conditions (e.g., patient experience).
>
> ---
> **W2: Case Study for Context Influences**
>
> Thanks for the constructive feedback! We will supplement the relevant case studies and analysis in the revised version.
>
> ---
> **W3: Flawed Reasoning in Comparison**
>
> Thank you! We have revised the corresponding part to avoid any misunderstanding. We would like to clarify that it is essential for a human physician to obtain sufficient information before making a diagnosis, and the same is true for LLMs. As shown in Figure 3, by increasing the number of dialogue turns, the amount of information collected and the diagnostic accuracy both increase. This suggests that current LLMs still have great potential to improve performance by collecting more information.
>
> ---
> **Q1: Expert Evaluation Correlation**
>
> For the subjective metrics, we believe that annotators without medical knowledge are better able to simulate the patient's perspective. For example, 'clarity' evaluates whether the communication is clear and understandable for those without medical jargon knowledge, while 'empathy' assesses whether a patient feels comfortable and reassured, even if they don't fully understand why the doctor inquires certain details. We have also invited expert-level annotators to ensure a more comprehensive evaluation. We will provide an update once the results are available.
>
> ---
> **Q2: Performance in Real Patients**
>
> While there remains a gap between the performance of these LLMs in simulated and real-world scenarios, we have made efforts to incorporate key features (see Table 1 and Figure 2) to bridge this gap, addressing various scenarios throughout the inquiry process (Diagnosis 9th ed. by Wan & Lu, 2018, pp. 77-78). Compared to previous works, we believe our approach brings us closer to real-world applicability.
> > Wan, X., & Lu, X. (2018). Diagnosis (9th ed.). People's Medical Publishing House. ISBN 978-7-117-26374-0.

---

### Official Review · Reviewer_R5PE · 2025-10-31

**Soundness:** 2
**Presentation:** 2
**Contribution:** 2
**Rating:** 4
**Confidence:** 4

**Summary:**

This paper introduces MAQUE, a benchmark for evaluating the inquiry skills of AI doctor agents. The authors argue that current benchmarks focus too much on final diagnostic accuracy and use patient simulators that are too simplistic.

MAQUE's first contribution is a dataset of 3,000 simulated patients. A key feature is "Atomic Information Units" (AIUs), which controls information disclosure to simulate a more realistic, passive patient. Its second contribution is a 5-dimensional evaluation framework. Instead of just accuracy, it also measures Inquiry Proficiency, Dialogue Competence, Inquiry Efficiency, and Patient Experience.

The authors tested several modern LLMs and found them to be "far from being effective physicians." Even top models fail to gather enough information, show little empathy, and struggle to balance accuracy with efficiency.

**Strengths:**

The paper addresses a clear gap. It rightly moves the focus from just diagnostic accuracy to the entire inquiry process and patient experience. This is a crucial step for building practical AI doctors.

The patient simulation method is a major strength. Its three-layer approach (Disclosure, Linguistics, Noise) creates a realistic testbed. However, using "Atomic Information Units" to control information flow is first proposed in MediQ (https://arxiv.org/abs/2406.00922), which the authors briefly cited in Table 1 but did not acknowledge for other high levels of similarity.

The 5-dimensional framework is thorough. It provides a holistic view of an agent's performance and captures overlooked aspects like patient communication (empathy, clarity) and inquiry skill (coverage, relevance).

**Weaknesses:**

* Oversimplification of Prior Work: The paper oversimplifies prior work. The claim (L066) that others ignore the "intermediate conversational process" is an overstatement. Existing work has already proposed fine-grained metrics (Schmidgall et al., 2024) and evaluated question-level quality (https://arxiv.org/abs/2502.14860). The paper fails to properly situate itself against this related work.

* Missing Related Work: The paper misses some related work. The suggestion (L133) to add more roles like a "triage nurse" overlooks research that has already explored this (e.g., https://arxiv.org/abs/2404.15155).

* The patients' personas and traits are not grounded in real-world clinical or psychological data. The categories for linguistic style, cognitive status, and emotional state seems arbitrary, and the characteristics seem randomly assigned.

* Heavy Reliance on LLM-as-judge: The paper uses an LLM (GPT-4o-Mini) to score subjective, human-centric metrics like "empathy," "clarity," and "coherence." This is a significant weakness as LLM-judges are known to have biases and may not be reliable proxies for true human experience. The validation in Appendix G even shows a weaker correlation for "clarity" (0.66) and uses non-medical experts, which is also a limitation.

* In the main experiment (Section 4.1), the paper separates the "inquirer" (the model being tested) from the "diagnostician" (a powerful, static GPT-5). This is an artificial setup. In a real-world agent, the ability to ask good questions is driven by its own internal diagnostic reasoning. By splitting them, the benchmark isn't testing an integrated agent, so the "inquiry" scores may not reflect real-world utility.

* Superficial Simulation of Human Behavior: While the simulation is a strength, its "noise injection" (Section 3.1, Appendix D.4) is still quite simplistic. Emotions and cognitive states (like "confused" or "anxious") are assigned and their intensity is just "predicted" each turn. A truly realistic simulation would have these states dynamically evolve based on the doctor's actions (e.g., a doctor using jargon might cause a patient to become "confused" or "frustrated").

* Minor Presentation Issues:
- Section 2.2 is verbose and could be condensed.
- L198 incorrectly states that Figure 2 contains an example patient prompt; it does not.

**Questions:**

1. How was the diversity of the 3,000 simulated patients (especially from "Patient-Zero") ensured? How faithful is this cohort to a realistic clinical distribution of demographics, condition prevalence, and severity?

2. How was the case distribution across 21 specialties determined? Was it based on data availability, or did it model a realistic case distribution for a general practitioner?

3. What is the justification for the five evaluation dimensions? The selection seems arbitrary. Why these specific metrics and not others?

4. How reliable is the LLM-as-a-judge for subjective metrics like "empathy"? Appendix G states validation used non-experts. How does using non-experts to validate a "patient" experience score affect its validity?

5. The paper notes (L409) that emotional responses can improve agent empathy. Is there a correlation between "Patient Experience" and "Task Success"? Does a more empathetic agent gather more information (higher "Coverage") and achieve a more accurate diagnosis?

---

> ### Author Response · Authors · 2025-11-24
>
> Thank you! Our point-by-point responses to your concerns are as follows:
>
> ---
> **W1: Citation of Related Work**
>
> Thank you for your helpful suggestions! We will discuss them in the revision and carefully improve the corresponding statements. Specifically, we have addressed the points with updates on lines 133, 150, and 210, refining the writing as you recommended. Nevertheless, As shown in the updated Table 1, MAQuE still demonstrates its comprehensive range of evaluation aspects, and we believe it can benefit the community.
>
> ---
> **W2: Patient Simulation**
>
> We use a 3 layer method to simulate patient agents(Figure2). For Layer 1, as noted in Line 219, patients often cannot provide complete information proactively. Previous works ([1], [2], [3]) highlight this characteristic, and we drew on methods similar to Fact Select ([3]) to simulate this behavior at a fine-grained level. For Layer 2 & 3, as mentioned in Line 162, patients exhibit various language styles, cognitive statuses, and emotional states in clinical environments ([3], [4]). Specifically, patients present features including frustration, anxiety, distraction, hostility, low educational levels etc., as outlined in Diagnosis ([5], pp. 77-78). We categorized these conditions to systematically simulate them.
> > [1] Samuel Schmidgall et al. Agentclinic: A multimodal agent benchmark to evaluate AI in simulated clinical environments. arXiv preprint arXiv:2405.07960, 2024.
> > [2] Zhihao Fan et al. AI hospital: Benchmarking large language models in a multi-agent medical interaction simulator. COLING, pp. 10183–10213, Abu Dhabi, UAE, 2024.
> > [3] Shuyue Stella Li et al. Mediq: Question-asking llms and a benchmark for reliable interactive clinical reasoning. NeurIPS 2024.
> >[4] Ruiyi Wang et al. Patient-ψ: Using large language models to simulate patients for training mental health professionals. EMNLP 2024, pages 12772–12797, Miami, Florida, USA.
> > [5] Huizi Yu et al. Simulated patient systems are intelligent when powered by large language model-based AI agents. arXiv:2409.18924, 2024.
> > [6] Wan, X., & Lu, X. (2018). Diagnosis (9th ed.). People's Medical Publishing House. ISBN 978-7-117-26374-0.
>
> ---
> **W3: LLM-as-judge Reliance**
>
> See Question 4.
>
> ---
> **W4: Inquiry-Diagnosis Relationship**
>
> As stated in Line 303, we uniquely focus on evaluating the inquiry capability of an LLM because a good diagnostician is not necessarily a good inquirer, and vice versa. This is demonstrated in Figure 4, which shows that domain-specific tuning improves diagnostic performance at the cost of inquiry capability. Additionally, our benchmark supports using a single LLM for both inquiry and diagnosis.
>
> ---
> **W5: Dynamics Simulation**
>
> We have incorporated this feature into our patient simulation (see Line 923), updating the patient's emotional state each turn based on the context so far. We will emphasize this point in the main text.
>
> ---
> **W6: Presentation Issues**
>
> Thank you! We have updated the manuscript accordingly based on your suggestions.

---

> ### Author Response · Authors · 2025-11-24
>
> **Q1:  Patient Diversity**
>
> Thank you for your thoughtful question. To ensure diversity in medical profiles, we utilized multiple data sources including different medical exams and medical journals(Line 201). Specifically, Patient-Zero was generated based on the medical knowledge base derived from Baidu Health Encyclopedia, with validated diversity in demographics, diseases, and exam results.
>
> ---
> **Q2: Specialty Distribution**
>
> As stated in Appendix A, the 21 specialties are categorized based on our collected patient profiles from diverse sources. We hope this feature helps support future research focusing on specific domains.
> The realistic distribution can be hard to estimate, as it varies with factors such as season and location.
>
> ---
> **Q3: Metrics Design**
>
> We design the five dimensions based on the inquiry guidance chapter from Diagnosis (9th ed.) by Wan and Lu(2018).
> - **Task Success** and **Efficiency** prioritize the AI Agent’s ability to successfully complete tasks and its operational efficiency.
> - **Information Proficiency** emphasizes that the completeness and accuracy of patient history significantly impact diagnosis and treatment (Diagnosis, p. 68).
> - **Dialog Competence** involves Adherence, "maintaining systematic, purposeful questioning, avoiding repetitive or disorganized inquiries"; Coherence, "adding smooth transitions without abrupt topic changes", "utilizing appropriate transition language". (Diagnosis, pp. 74-75)
> - **Patient Experience** focuses on Clarity, "avoiding the confusion caused by a series of rapid-fire questions" and "minimizing technical jargon"; Empathy, "creating a supportive and warm environment", "making patients feel heard and valued", "using praise and encouraging language". (Diagnosis, pp. 74-75)
> > Wan, X., & Lu, X. (2018). Diagnosis (9th ed.). People's Medical Publishing House. ISBN 978-7-117-26374-0.
>
> ---
> **Q4: Non-Expert Annotator**
>
> For the subjective metrics, we believe that annotators without medical knowledge are better able to simulate the patient's perspective. For example, 'clarity' evaluates whether the communication is clear and understandable for those without medical jargon knowledge, while 'empathy' assesses whether a patient feels comfortable and reassured, even if they don't fully understand why the doctor inquires certain details. We have also invited expert-level annotators to ensure a more comprehensive evaluation, and will provide an update once the results are available.
>
> ---
> **Q5: Empathy v.s. Coverage**
>
> We do not observe that improved empathy necessarily leads to better task success or inquiry quality. For instance, UltraMedical demonstrated notably higher empathy scores, yet no significant improvements were observed in other evaluation metrics. We believe existing models struggle to balance these metrics effectively. This is demonstrated by GPT-4o and Qwen3-8B, which achieve the best performance on task success but perform relatively worse on empathy in the closed-source and open-source categories, respectively.

---

### Official Review · Reviewer_FMAp · 2025-11-04

**Soundness:** 2
**Presentation:** 2
**Contribution:** 2
**Rating:** 2
**Confidence:** 3

**Summary:**

The paper introduces MAQUE, a benchmark to evaluate medical multi-turn inquiry abilities of LLM-based medical agents. MAQUE include human-like behavioral such as disclosure control, linguistic variation, and noise injection. The benchmark assesses multiple complementary dimensions with various fine-grained metrics. Experiments across frontier, open-source, and medical-specific models reveal model weaknesses, behavioral sensitivity, and trade-offs between diagnostic accuracy and inquiry cost.

**Strengths:**

1. MAQUE introduces a three-layer simulation for patient behavior with human-like noise behavioral injection. This aligns with reality.
2. The benchmark is built on a large scale and covers 21 medical departments. Its multi-dimensional evaluation provides a comprehensive view of conversational quality.
3. The experiment tests multiple competitive models across closed, open-source, and medical domain-specific. Inquiry strategies evaluation is interesting as it reveals failures of CoT and self-consistency on this benchmark.

**Weaknesses:**

1. The primary results presented in Table 2 - 4 are all point estimates. The paper does not report any measures of variance across patient agents and random seeds. This is a critical omission, as it is impossible to determine if the performance differences between models are statistically significant or merely noise.
2. Many core metrics, e.g. Patient Experience and Dialogue Competence are evaluated using an LLM-as-judge. The human validation study does not report the IAA among annotators. Without  inter annotator agreement, it's difficult to assess the reliability of their judgments or the meaning of the correlation scores (e.g. the low 0.6635 for Clarity ). Also, the human validation is conducted with non-expert and synthetic conversation, no validation with clinical staff or real patient interactions.
3. Task Success is dependent on a single, unevaluated GPT-5 used as a fixed "diagnostic agent". The paper assumes this model is "powerful" but presents no independent evidence of its diagnostic accuracy or robustness across the different medical specialties.

**Questions:**

1. Is there any analysis testing the robustness of the LLM-as-judge itself? For example, does the judge's scoring for metrics like clarity or empathy remain stable when the only variable changed is the patient's linguistic style or emotional state?
2. Non-expert annotators is a significant concern for a medical conversation benchmark. Do you have any plans to calibrate these subjective metrics and annotations from clinical experts or physicians?

---

> ### Author Response · Authors · 2025-11-24
>
> Thank you! Our point-by-point responses to your concerns are as follows:
>
> ---
> **W1: Variance Measures**
>
> Please note that MAQuE is much larger than previous benchmarks, which allows it to provide more stable results. To address your concerns, we have conducted three runs with different random seeds. While there are minor variations in the results, the ranking of the models remains consistent, which reinforces the reliability of our conclusions.
>
> | Inquiry Model   | Acc.       | Rob.     | Cov.| Rel.| Adh.| Coh.| #Ques.| #Tok.(k.)| Clar.        | Emp.|
> |-----------------|----------------|----------------|----------------|----------------|----------------|----------------|----------------|----------------|----------------|----------------|
> | GPT-4o          | 0.683 ± 0.006  | 0.866 ± 0.006  | 0.370 ± 0.004  | 0.902 ± 0.009  | 0.973 ± 0.008  | 0.858 ± 0.026  | 9.673 ± 0.031  | 0.236 ± 0.037  | 0.805 ± 0.010  | 0.540 ± 0.013  |
> | GPT-5-Chat      | 0.680 ± 0.013  | 0.866 ± 0.007  | 0.304 ± 0.001  | 0.904 ± 0.010  | 0.988 ± 0.004  | 0.837 ± 0.024  | 8.850 ± 0.130  | 0.202 ± 0.009  | 0.733 ± 0.033  | 0.486 ± 0.028  |
> | Qwen3-8B        | 0.629 ± 0.016  | 0.859 ± 0.009  | 0.325 ± 0.003  | 0.900 ± 0.008  | 0.959 ± 0.008  | 0.768 ± 0.029  | 9.929 ± 0.013  | 0.258 ± 0.016  | 0.666 ± 0.034  | 0.421 ± 0.014  |
> | Baichuan-M2-32B | 0.594 ± 0.017  | 0.822 ± 0.007  | 0.342 ± 0.003  | 0.921 ± 0.005  | 0.965 ± 0.003  | 0.798 ± 0.025  | 9.924 ± 0.027  | 0.353 ± 0.018  | 0.669 ± 0.032  | 0.472 ± 0.027  |
>
>
> ---
> **W2: Human Annotation Validation**
>
> *2.1 Inter annotator alignment*
>
> We use two other judge models to compute the Pearson correlation. The results are as follows and will be updated in the revision.
> | Metric    | Judge_X   | Judge_Z  | r (Pearson) | p-value |
> | --------- | --------- | -------- | ----------- | ----------- |
> | Adherence | GPT-4o-Mini | GPT-5-Chat | 0.9974      | 0.0026      |
> | Adherence | GPT-4o-Mini | Gemini-2.5-Flash| 0.9903      | 0.0097      |
> | Coherence | GPT-4o-Mini | GPT-5-Chat| 0.9504      | 0.0496      |
> | Coherence | GPT-4o-Mini | Gemini-2.5-Flash| 0.9822      | 0.0178      |
> | Clarity   | GPT-4o-Mini | GPT-5-Chat| 0.8394      | 0.1606      |
> | Clarity   | GPT-4o-Mini | Gemini-2.5-Flash| 0.8514      | 0.1486      |
> | Empathy   | GPT-4o-Mini | GPT-5-Chat| 0.9986      | 0.0014      |
> | Empathy   | GPT-4o-Mini | Gemini-2.5-Flash   | 0.9912      | 0.0088      |
>
> *2.2 Clinical staff validation*
>
> We believe that annotators without medical knowledge are better able to simulate the patient's perspective. For example, "clarity" evaluates whether the communication is clear and understandable for those without medical jargon knowledge, while "empathy" assesses whether a patient feels comfortable and reassured, even if they don't fully understand why the doctor inquires certain details. We have also invited expert-level annotators to ensure a more comprehensive evaluation, and will provide an update once the results are available.
>
> *2.3 Real patient interactions*
>
> As stated in Line 130 and 148, previous work has shown that LLMs can effectively simulate patients. This method provides better control and ensures replicability without the associated ethical concerns. Besides, we focus on building a simulated environment to efficiently evaluate an AI doctor. While involving real patients is valuable, it is also costly and time-consuming, placing them beyond the scope of this study.

---

> ### Author Response · Authors · 2025-11-24
>
> **W3: Capability of Diagnostic Agent**
>
> To the best of our knowledge, GPT-5 demonstrates top-tier performance in medical diagnosis among LLMs, even surpassing human experts by 13.30% in medical text reasoning and understanding [1].
> > [1] Shansong Wang, Mingzhe Hu, et al. "Capabilities of GPT-5 on Multimodal Medical Reasoning," arXiv:2508.08224 [cs.CL].
>
> Also, we added a set of experiments where we cross-combine different inquiry and diagnosis models to observe variations in the results:
>
> | Diagnosis model \ Inquiry model | GPT-5 | GPT-5-Chat | GPT-4o | Qwen3-8B | Baichuan-M2-32B |
> | ------------------------------- | ----- | ---------- | ------ | -------- | --------------- |
> | GPT-5                           | 0.54  | 0.67       | 0.66   | 0.62     | 0.59            |
> | GPT-5-Chat                      | 0.55  | 0.62       | 0.61   | 0.56     | 0.57            |
> | GPT-4o                          | 0.43  | 0.46       | 0.36   | 0.32     | 0.38            |
> | Qwen3-8B                        | 0.39  | 0.46       | 0.38   | 0.42     | 0.4             |
> | Baichuan-M2-32B                 | 0.48  | 0.54       | 0.46   | 0.47     | 0.44            |
>
> - The ranking of inquiry performance is consistent with the trend observed in Table 2.
> - GPT-5 consistently demonstrates the best diagnostic performance, supporting its suitability as a stable diagnostic agent, while its inquiry performance is not the highest, indicating that its reasoning module is not well-suited for the inquiry task.
>
> ---
> **Q1: Robustness of LLM-as-Judge**
>
> Thank you for your insightful question. As stated in Line 88, LLM-as-a-Judge is widely used for evaluating various conversations, thus we believe it is robust to such variations. Besides, the presented results were tested using consistent simulated patients, which ensures our conclusions are not affected by this concern.
>
> ---
> **Q2: Non-expert Annotators**
>
> See W2.

---

### Note · Authors · 2025-12-02

I have read and agree with the venue's withdrawal policy on behalf of myself and my co-authors.